# Ratchet-free solid-state inertial rotation of a guest ball in a tight tubular host

Taisuke Matsuno [1,2], Yusuke Nakai [3], Sota Sato [1,2], Yutaka Maniwa[3] & Hiroyuki Isobe [1,2]

Dynamics of molecules in the solid state holds promise for connecting molecular behaviors with properties of bulk materials. Solid-state dynamics of [60]fullerene ($C_{60}$) is controlled by intimate intermolecular contacts and results in restricted motions of a ratchet phase at low temperatures. Manipulation of the solid-state dynamics of fullerene molecules is thus an interesting yet challenging problem. Here we show that a tubular host for $C_{60}$ liberates the solid-state dynamics of the guest from the motional restrictions. Although the intermolecular contacts between the host and $C_{60}$ were present to enable a tight association with a large energy gain of –14 kcal mol$^{-1}$, the dynamic rotations of $C_{60}$ were simultaneously enabled by a small energy barrier of +2 kcal mol$^{-1}$ for the reorientation. The solid-state rotational motions reached a non-Brownian, inertial regime with an extremely rapid rotational frequency of 213 GHz at 335 K.

[1] Department of Chemistry, The University of Tokyo, Hongo 7-3-1, Bunkyo-ku, Tokyo 113-0033, Japan. [2] JST, ERATO, Isobe Degenerate π-Integration Project, Hongo 7-3-1, Bunkyo-ku, Tokyo 113-0033, Japan. [3] Department of Physics, Tokyo Metropolitan University, Hachioji, Tokyo 192-0397, Japan. These authors contributed equally: Taisuke Matsuno, Yusuke Nakai. Correspondence and requests for materials should be addressed to H.I. (email: isobe@chem.s.u-tokyo.ac.jp)

Dynamics of molecules in the solid state holds promise for connecting events at the molecular size with properties of larger, bulk materials[1], although the solid-state dynamics of molecules are severely restricted by intermolecular contacts[2]. A soccer-ball-shaped molecule, [60]fullerene ($C_{60}$), is known for its unique dynamic characteristics[3–5], and, in particular, a peculiar solid-state dynamics has been discovered. In the solid state, the $C_{60}$ molecules dynamically rotate despite intimate intermolecular contacts involved therein[6, 7]. The rotational motions, however, are not completely free from motional restrictions, and the dynamics is under the influence of the 32-faced polyhedral shape. Between the two dynamic phases observed with the $C_{60}$ solid, the restricted ratchet phase emerges from the face-to-face contacts of $C_{60}$ molecules below 260 K[6–8]. Above the phase transition temperature, the other rotator phase emerges to allow for the rapid rotations of $C_{60}$ molecules. In this high-temperature region, the rotational motions approach the boundary of the diffusional, Brownian rotations with a motional measure of $\chi = 2.4$ at 331 K (see below). Manipulation of such unique solid-state dynamics is a challenging yet interesting subject to be explored for dynamic solid-state materials[8, 9], and an interesting method for the dynamics control has been exploited by using carbon nanotubes (CNT)[10]. The dynamics of $C_{60}$ has thus been investigated in the supramolecular composites, so-called CNT peapods. Although changes in the dynamic motions of encapsulated $C_{60}$ molecules have been suggested[11], an inhomogeneous nature intrinsic to the CNT materials hampered reproducible measurements as well as definitive, clear-cut conclusions[11, 12]. We have recently introduced a molecular peapod with a finite segment of helical CNT, i.e., [4]cyclo-2,8-chrysenylene ([4]CC)[13, 14], and started investigating the physical characteristics of these molecular entities possessing discrete tubular structures (Fig. 1a)[15]. The rigid host of (12,8)-[4]CC tightly encapsulates $C_{60}$ in its inner space with the highest association constants ever recorded with $C_{60}$ ($K_a \sim 10^{12}$ M$^{-1}$ and $\Delta H \sim -14$ kcal mol$^{-1}$ in benzene)[15, 16], and despite such tight associations, the dynamic rotational motions of the $C_{60}$ guest are present both in solution[15] and solid[17].

Here we report a complete physical picture of solid-state dynamics of $C_{60}$ in the tubular host. Anomalous effects of the tubular host on the rotational dynamics of the guest have been revealed. Albeit paradoxically, a tight association and a low friction are concurrently achieved. This study should stimulate future developments of unique dynamic supramolecular systems assembled solely by van der Waals interactions[18].

## Results

**Crystallography.** The solid-state dynamic motions were first indicated by variable-temperature (VT) crystallographic analyses. Crystals of single-handed, (P)-(12,8)-[4]CC⊃$C_{60}$ were grown from a methanol/dichloromethane solution, and six single crystals were obtained from an identical batch. Under six different temperatures applied to each crystal, the crystals were subjected to diffraction analysis with a synchrotron X-ray beam (PF-AR NE3A/KEK Photon Factory). Six independent diffraction datasets were converged, respectively, into six molecular structures of (P)-(12,8)-[4]CC⊃$C_{60}$. Each structure was finalized with four disordered $C_{60}$ orientations, and the temperature-dependent fluctuations of $C_{60}$ orientations were also visualized by raw electron density maps with $2F_o$–$F_c$[19]. A complete set of the crystal data is summarized in Supplementary Fig. 1, and representative data are shown in Fig. 1. The molecular structure of (P)-(12,8)-[4]CC⊃$C_{60}$ shown in Fig. 1b was essentially a mirror image of that of an enantiomer, (M)-(12,8)-[4]CC⊃$C_{60}$, determined in our previous study[17]. Independent of the temperatures (95, 140, 180, 220, 260, and 295 K), the disordered $C_{60}$ molecules (24 molecules in total

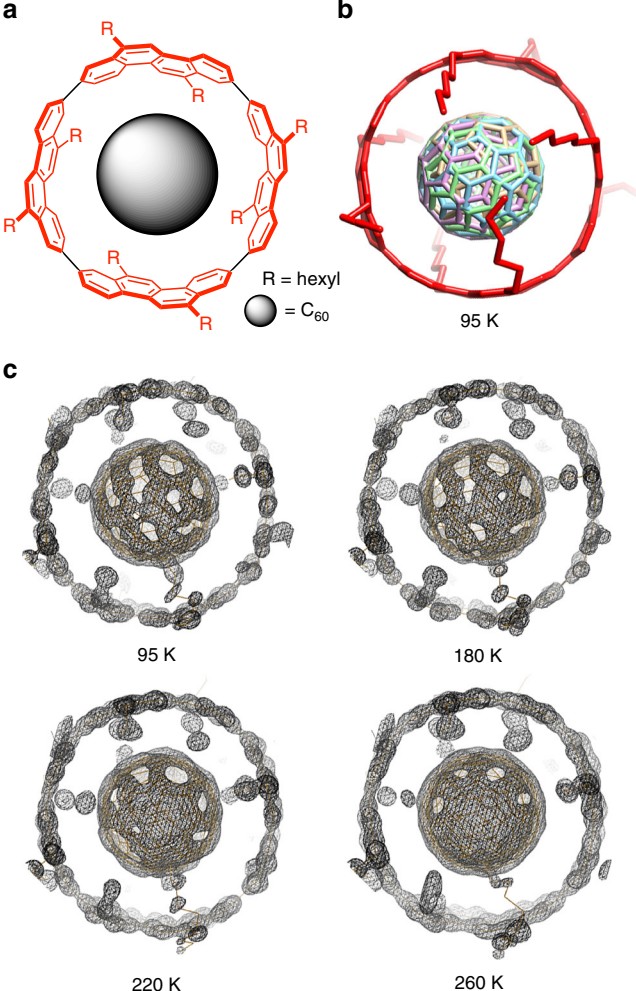

**Fig. 1** Variable-temperature crystallographic analyses of (P)-(12,8)-[4]CC⊃$C_{60}$. **a** Molecular structure shown in the chemical diagrams. **b** A crystal structure at 95 K, shown in tube models. Four disordered $C_{60}$ orientations are shown in different colors. Disordered alkyl chains and hydrogen atoms are omitted for clarity. **c** Temperature-dependent electron density mappings with $2F_o$–$F_c$ (RMSD: 1.5$\sigma$) at 95, 180, 220, and 260 K

with six crystal structures having four $C_{60}$ orientations) shared a common center of mass that was located at the center of the [4]CC tube (Supplementary Fig. 1). Although the effects of temperature were not clear merely by examining these molecular structures, the contour maps of electron densities (root-mean-square deviation, RMSD: 1.5$\sigma$) clarified the temperature effects on the molecular orientations of $C_{60}$ in the host. As shown in Fig. 1c, the distributions of electrons at low temperatures (e.g. 95 and 180 K) showed biased locations of carbon atoms, indicating the presence of favorable low-energy orientations. At higher temperatures such as 220 and 260 K, the vacant spaces not distributed with electrons diminished, and the evenly distributed, ball-shaped electron mappings of $C_{60}$ emerged. This result indicates that the unfavorable orientations of $C_{60}$ are energetically separated by minute gaps from the favorable orientations and that the $C_{60}$ orientations increase the degree of freedom at high temperatures.

**Nuclear magnetic resonance spectra.** The solid-state dynamics of (P)-(12,8)-[4]CC⊃$C_{60}$ were next quantitatively investigated by VT nuclear magnetic resonance (NMR) spectroscopy. Static solid-state $^{13}$C NMR spectra under a 9.39-T magnetic field are shown in Fig. 2. As was recorded with the enantiomer[17], a narrow

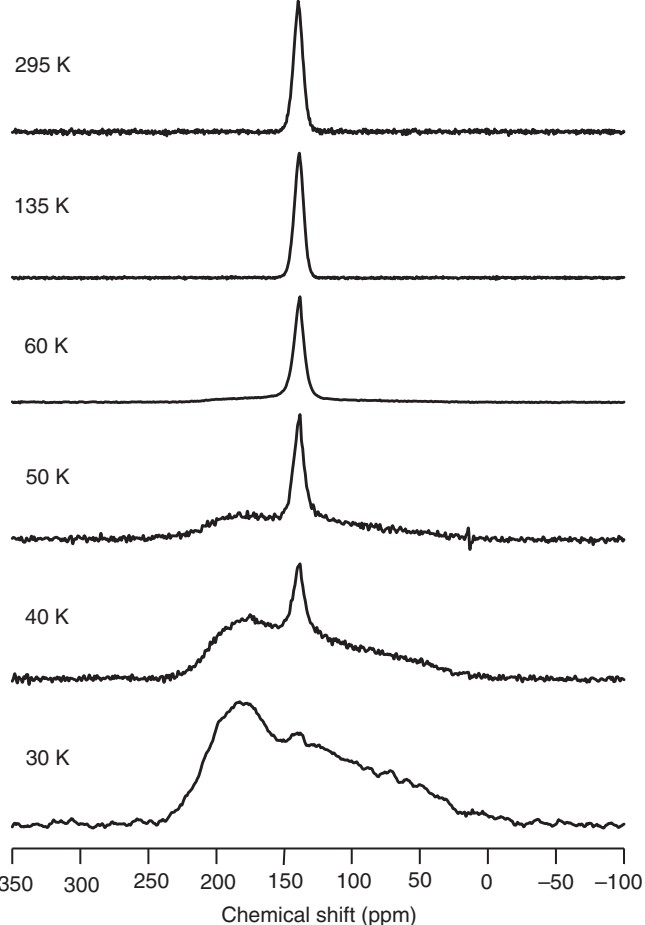

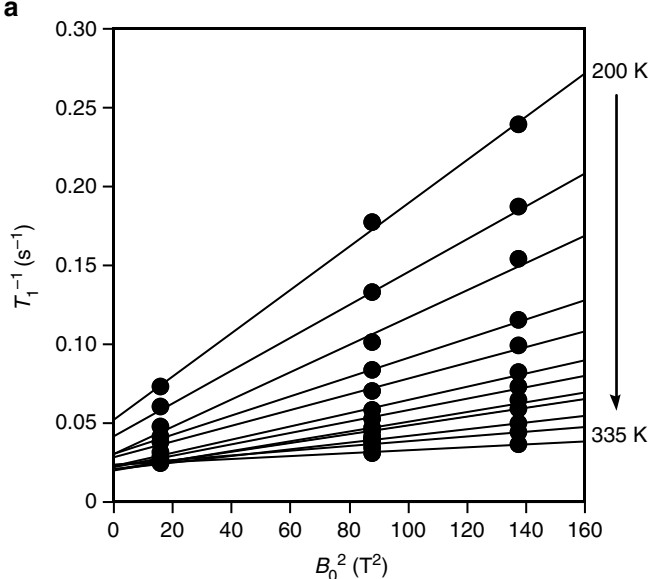

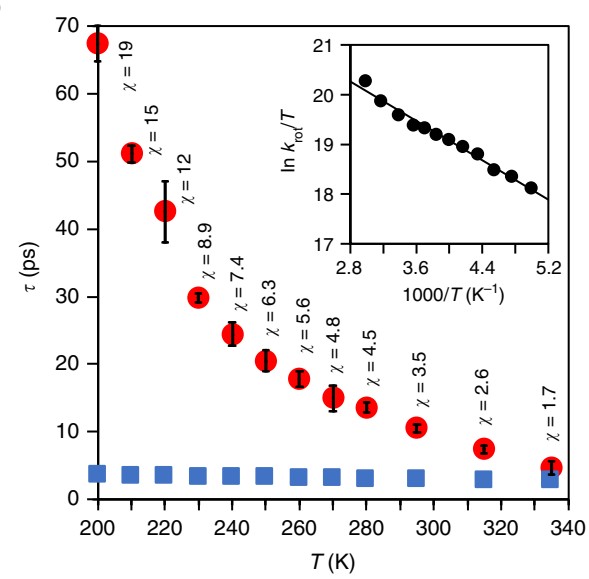

**Fig. 2** Variable-temperature solid-state NMR analysis of $(P)$-(12,8)-[4]CC⊃C$_{60}$. Measurements were performed under static conditions (9.39 T) without MAS. Fullerene C$_{60}$ was enriched with $^{13}$C (20–30%)

symmetric peak originating from C$_{60}$ (20–30% $^{13}$C-enriched) was recorded under static conditions without magic-angle spinning (MAS) at 295 K. Previously, the symmetric peak was observed to be unchanged down to 243 K, that is, the lowest temperature of our conventional spectrometer[17]. In the present study adopting home-made NMR instruments[11], the temperature was further lowered to 30 K, and below 50 K, the symmetric peak with averaging effects from the molecular motions disappeared to show a powder pattern originating from non-averaged chemical shift anisotropy (CSA). The powder pattern of $(P)$-(12,8)-[4]CC⊃C$_{60}$ at 50 K was similar to that of intact C$_{60}$ at 143 K under an identical magnetic field[7]. This temperature difference indicated that in the presence of the tight tubular host, the speed of C$_{60}$ rotations becomes slow compared to the CSA width at a much lower temperature than that of intact C$_{60}$ (50 K vs. 143 K).

**Rotational frequency**. Quantitative kinetic analyses were carried out through measurements of spin-lattice relaxation time ($T_1$)[7]. In short, to exclude the field-independent non-CSA (NCSA) contributions such as magnetic dipole−dipole relaxation (C−C/C−H), the $T_1$ values of C$_{60}$ in [4]CC were recorded under three different magnetic fields ($B_0 = 4.00$, 9.39, and 11.7 T) (Supplementary Figs. 2–4). The field-dependent $T_1$ values were then plotted against $B_0^2$ to determine the $\tau$ values from the slope of the linear correlations (Fig. 3a)[20]. A series of measurements in the temperature range of 200–335 K allowed for a plot of the

**Fig. 3** Rotational dynamics revealed from spin-lattice relaxation time ($T_1$) measurements. **a** Field-dependent $T_1$ values for the temperature range of 200–335 K under three different magnetic fields (4.00, 9.39, and 11.7 T). A linear correlation was visualized by plotting the reciprocal $T_1$ against the square of the magnetic field $B_0^2$, and the rotational correlational time ($\tau$) was obtained from the slope. **b** Temperature-dependent $\tau$ values. Experimental $\tau$ values are shown in red circles, and theoretical natural-limit values, $\tau_{FR}$, are shown in blue squares. The bars show the standard errors of the estimate. The dynamics measures of $\chi$ ($=\tau/\tau_{FR}$) are shown, and the smallest value, 1.7, revealed the presence of inertial rotational motions. The inset shows the Eyring plot adopting $k_{rot}$ ($=1/\tau$) to disclose the energetics for the rotations

temperature-dependent $\tau$ values as shown in Fig. 3b. At the highest temperature, 335 K, the smallest value of the rotational correlation time with a standard error of the estimate was recorded as $\tau = 4.7 \pm 1.0$ ps, which was smaller than that observed with intact C$_{60}$ ($\tau = 6.8$ ps at 331 K)[6]. The rotational correlation time was converted to the rotational frequency that reached the largest value of $k_{rot} = 213 \pm 45$ GHz at 335 K to show the presence of an extremely fast solid-state rotational motions of C$_{60}$ in the tubular host.

**Energetics of rotations**. Comparisons of $C_{60}$ dynamics in the tubular host with those of intact, solid $C_{60}$ revealed unique roles of the [4]CC tube[7]. As shown in Fig. 3, the $\tau$ value of $C_{60}$ in (P)-(12,8)-[4]CC showed one monotonic exponential decay with temperature throughout the investigated temperature range (200–335 K). In contrast, previous studies of intact $C_{60}$ have shown that the phase transition of ratchet/rotor motions is present at 260 K, dividing two different exponential decays of $\tau$ values (see also Supplementary Fig. 5)[6–8]. The single exponential decay observed with (P)-(12,8)-[4]CC⊃$C_{60}$ was similar to that of the high-temperature region of intact $C_{60}$, which indicated the presence of ratchet-free, rotor motions throughout the temperature range. The single exponential decay of $\tau$ values was further analyzed by the Eyring plot ($1/T-\ln(k_{rot}/T)$; Fig. 3, inset) to elucidate the energy barriers of the $C_{60}$ rotations in the host as $\Delta G^{\ddagger} = +2.45 \pm 0.13$ kcal mol$^{-1}$ (335 K), $\Delta H^{\ddagger} = +1.96 \pm 0.08$ kcal mol$^{-1}$ and $\Delta S^{\ddagger} = -1.46 \pm 0.32$ cal mol$^{-1}$ K$^{-1}$. The energy barriers showed the origins of smooth motions, albeit seemingly paradoxical, in the presence of the large association energy[15, 21]. The inner surfaces of [4]CC were smoothly curved without inflection lines (Supplementary Fig. 6)[17], which should structurally eliminate face-to-face intermolecular contacts that generated the ratchet, restricted motions of $C_{60}$[7].

**Inertial rotation**. The smallest correlation time of $C_{60}$ rotations, $\tau = 4.7$ ps, recorded with (P)-(12,8)-[4]CC⊃$C_{60}$ at the highest temperature (335 K) showed the presence of unique dynamic motions. As has been reported with intact $C_{60}$[7], the modes of rotational motions can be elucidated by comparing the experimental correlation time ($\tau$) with the natural-limit value for free rotations ($\tau_{FR}$) with the measure of $\tau/\tau_{FR}$ ($\equiv\chi$) (see also Methods for details)[22, 23]. Thus, according to Steele's theoretical proposal[22, 23], the $\chi$ value of 2.4 recorded with intact $C_{60}$ at 331 K led Johnson to conclude the presence of diffusional $C_{60}$ rotations near the boundary of the inertial regime ($\chi < 2$). The theoretical $\tau_{FR}$ value of $C_{60}$ at 335 K was calculated as 2.7 ps (see also Methods)[7], and the smallest $\tau$ value, $4.7 \pm 1.0$ ps, of (P)-(12,8)-[4]CC⊃$C_{60}$ was thus converted to the small motional measure, $\chi = 1.7 \pm 0.4$. The $\chi$ measure below 2 showed that the rotational motions in the [4]CC host reached the inertial regime in the solid state.

## Discussion

The structural investigations of $C_{60}$ dynamics in a tubular supramolecular host revealed the presence of unique rotational motions, which reached the inertial rotational regime at 335 K. The ratchet-free rotational motions took place in smooth chiral environments provided by helically arranged $sp^2$-carbons[17, 24], and exploration of the chirality-related dynamics under the control of classical mechanics should be of great interest for future studies[25]. Investigations of energy inputs other than thermal energies should also expand the scope of the unique molecular bearings[21, 26].

## Methods

**Materials**. The tubular molecule, (P)-(12,8)-[4]CC, was converted to the molecular peapod, (P)-(12,8)-[4]CC ⊃ $C_{60}$, by encapsulating $^{13}$C-enriched $C_{60}$ (20–30%, MER Corporation) in solution[13, 17]. Thus, in CD$_2$Cl$_2$ (2.0 mL), (P)-(12,8)-[4]CC (27.1 mg, 17.2 μmol) was mixed with a slightly excess amount of $C_{60}$ (13.0 mg, ca. 18 μmol), and the mixture was sonicated for 30 min. An excess amount of $C_{60}$ remained insoluble, and its solid was removed by filtration. The formation of 1:1 complex in solution was confirmed by solution-phase NMR analyses. The solid specimens of the complex were obtained by removing the solvent and were used for the solid-state analyses.

**VT crystallographic analyses**. Six single crystals of (P)-(12,8)-[4]CC⊃$C_{60}$ were obtained from a methanol/dichloromethane (ca. 1:1 v/v) solution at 3 °C. A single

crystal was mounted on a thin polymer tip with cryoprotectant oil. The diffraction analyses with synchrotron X-ray sources were conducted, respectively, at 95, 140, 180, 220, 260, and 295 K at beamline PF-AR NE3A with the Dectris PILATUS 2M-F PAD detectors at the KEK Photon Factory. Temperature was controlled by the cooling device developed in KEK Photon Factory with dry nitrogen gas flow. The diffraction data were processed with the XDS software program[27]. The structure was solved by direct method[28] and refined by full-matrix least-squares on $F^2$ using the SHELXL-2014/7 program suite[29] running with the Yadokari-XG 2009 software program[30]. In the refinements, fullerene molecules were treated as four rigid body models and restrained by SIMU, alkyl groups were partially restrained by SIMU, DFIX, and DANG, and diffused solvent molecules were treated by SWAT. Twinning was treated with TWIN/BASF instructions. The non-hydrogen atoms were analyzed anisotropically, and hydrogen atoms were input at the calculated positions and refined with a riding model. Electron density mapping was performed on a COOT software program[31]. The Hirshfeld surface analyses[32] were performed using the CrystalExplorer software program[33]. The refinement data are shown in Supplementary Tables 1–6.

**NMR measurements**. Three different NMR instruments were used for the VT and field-dependent analyses. The magnetic fields are 4.00, 9.39, and 11.7 T (resonance frequency of $^{13}$C = 42.9, 101, and 125 MHz). The 4.00- and 9.39-T instruments were assembled in-house and were fully equipped for ultralow-temperature measurements. The 11.7-T instrument was commercially available products (JEOL ECA 500). The $^{13}$C NMR spectra of the solid specimen were obtained under 9.39 T in a temperature range of 30–295 K without applying MAS. The spin-lattice relaxation time ($T_1$) was measured by using the saturation-recovery method, and its temperature dependency was tracked in a range of 200–335 K. The magnetic field dependency of $T_1$ was traced under 4.00, 9.39, and 11.7 T. The $T_1$ data are shown in Supplementary Figs. 2–4.

**Determination of $\tau$ values**. The $T_1$ values were then converted to $\tau$ values by a method reported in the literature[6]. In short, the $T_1$ value is composed of a CSA part ($T_{1CSA}$) and an NCSA part ($T_{1NCSA}$) in the form of

$$\frac{1}{T_1} = \frac{1}{T_{1CSA}} + \frac{1}{T_{1NCSA}}. \tag{1}$$

The $T_{1CSA}$ part depends on the external magnetic field ($B_0$) in the form of

$$\frac{1}{T_{1CSA}} = B_0^2\gamma^2\left(2A^2\frac{\tau}{1+9\omega^2\tau^2} + \frac{2}{15}S^2\frac{\tau}{1+\omega^2\tau^2}\right), \tag{2}$$

where $\gamma$ is the $^{13}$C magnetogyric ratio ($67.31\times10^6$ rad s$^{-1}$ T$^{-1}$), $\omega$ is the angular Larmor frequency ($=\gamma B_0$) and $A^2$ and $S^2$ factors are derived from antisymmetric and symmetric components of the shielding tensors. The $S^2$ factor was calculated from the values obtained by the simulation of the $^{13}$C powder pattern under 9.4 T at 30 K, and the $A^2$ factor was adopted from the value of intact $C_{60}$[6]. According to this equation, we determined the $\tau$ values by using $1/T_1$-$B_0^2$ plots.

**Classic mechanics calculations of $\tau_{FR}$**. The natural-limit rotational correlation time for free rotation ($\tau_{FR}$) was calculated by a method reported in the literature[6]. Thus, the moment of inertia ($I$) of a hollow carbon-shell ball of $C_{60}$ is $1.0\times10^{-43}$ kg m$^2$, and the natural-limit $\tau_{FR}$ is calculated with

$$\tau_{FR} = \frac{3}{5}\sqrt{\frac{I}{k_B T}} \tag{3}$$

where $k_B$ is the Boltzmann constant and $T$ is the temperature.

**Data availability**. Crystallographic data are available at Cambridge Crystallographic Database Centre (https://www.ccdc.cam.ac.uk) as CCDC1821725, 1821726, 1821727, 1821728, 1821729, and 1821730. All other data that support the findings of this study are available from the corresponding author upon reasonable request.

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

## Acknowledgements
We thank Prof. B.I. Halperin (Harvard) for his important suggestions, S. Takahashi and M. Oinuma (ERATO) for the preparation of [4]CC and Central Glass Co. for the gift of hexafluoroisopropanol. We were granted access to the X-ray diffraction instruments of KEK Photon Factory (no. 2017G082) and to the NMR instrument of NIMS micro-structural characterization platform (MEXT Nanotechnology Platform). This study is partly supported by JST ERATO (JPMJER1301) and KAKENHI (17H01033, 16K05681, 16K04864, 25102007).

## Author contributions
H.I. launched the research project. T.M. and S.S. performed the crystallographic studies, and T.M., Y.N., S.S., and Y.M. carried out the solid-state NMR investigations. All authors analyzed and discussed the results, and T.M. and H.I. wrote the manuscript.

## Additional information

**Competing interests:** The authors declare no competing interests.

