## [Peer Review File · Nature Communications]

REVIEWERS' COMMENTS:

Reviewer #1 (Remarks to the Author):

This is a very interesting manuscript by Isobe and coworkers that follows the already beautiful work done studying the supramolecular complex of C60 and a finite segment of a carbon nanotube. In previous work, it was shown that these finite fragments of CNTs synthesized by Isobe and coworkers had the strongest association constants of any structure for C60. What was not known is an absolute value for the rotational barrier of C60 in these complexes. At standard temperatures available to solid state NMR studies (243K), it had been previously reported that C60 was rotating rapidly on the NMR timescale. In this work, the authors use some home-made NMR equipment to now record NMR spectra down to 30K. Using some fairly sophisticated analysis that is beyond my expertise, they are able to extract a activation barrier of around 2 kcal/mol for rotation of C60. They also show at higher temperatures (335K), the rotational frequency reaches 213 GHz, which is remarkable. Although some of the points such as that about the inertial regime of rotation stretch beyond my expertise, I find the work to be very fascinating. The fact that a molecule can have such a strong binding affinity, yet allow for very fast movement in the solid state is completely unintuitive. I would highly recommend publishing this work in Nature Communications. Many people will enjoy reading about the rotational dynamics in the solid state of this unusual carbonaceous host/guest complex.

Reviewer #2 (Remarks to the Author):

This manuscript by Isobe and coworkers shows that a C60 molecule rotates very rapidly, reaching a non-Brownian inertial regime at 335 K, when it is bound inside a molecular nanotube. This result is fascinating because the high binding energy of the C60 for the tubular host might have been expected to lead to a high barrier to rotation. The findings from this study will attract the interest of scientists from a wide range of disciplines and the work is significant enough to justify publication in Nature Communications.

The concept behind this study is similar to that of reference 16 (a PNAS paper from the same team), but this new study goes far beyond the previous one. The measurement of rotational dynamics over a wide range of temperatures is a significant advance.

The manuscript is well written and highly readable. It is suitable for publication with minor revision.

When the authors introduce the [4]cyclo-2,8-chrysenylene molecular nanotube host on page 3, it would be worth pointing out that this compound is a single enantiomer, otherwise readers may assume that it is racemic, then become confused by the discussion of chirality later in the article.

When the authors state that the rotation frequency of C60 in the host at 335 K is 213 GHz they should state the rotation frequency of C60 in pure C60 at the same temperature, so as to place this value in context.

One very minor suggestion for rewording on page 4, line 85: Change "minute, small" to "minute" or "small".

Point-by-point response (Our responses are shown in blue.)

REVIEWERS' COMMENTS:

Reviewer #1 (Remarks to the Author):

This is a very interesting manuscript by Isobe and coworkers that follows the already beautiful work done studying the supramolecular complex of C60 and a finite segment of a carbon nanotube. In previous work, it was shown that these finite fragments of CNTs synthesized by Isobe and coworkers had the strongest association constants of any structure for C60. What was not known is an absolute value for the rotational barrier of C60 in these complexes. At standard temperatures available to solid state NMR studies (243K), it had been previously reported that C60 was rotating rapidly on the NMR timescale. In this work, the authors use some home-made NMR equipment to now record NMR spectra down to 30K. Using some fairly sophisticated analysis that is beyond my expertise, they are able to extract a activation barrier of around 2 kcal/mol for rotation of C60. They also show at higher temperatures (335K), the rotational frequency reaches 213 GHz, which is remarkable. Although some of the points such as that about the inertial regime of rotation stretch beyond my expertise, I find the work to be very fascinating. The fact that a molecule can have such a strong binding affinity, yet allow for very fast movement in the solid state is completely unintuitive. I would highly recommend publishing this work in Nature Communications. Many people will enjoy reading about the rotational dynamics in the solid state of this unusual carbonaceous host/guest complex.

We would like to thank this reviewer for his/her time spared for evaluation and also for his/her favourable evaluation. We are pleased to learn that this reviewer shares the excitement on this unexpected discovery of fast rotations in the tight host.

Reviewer #2 (Remarks to the Author):

This manuscript by Isobe and coworkers shows that a C60 molecule rotates very rapidly, reaching a non-Brownian inertial regime at 335 K, when it is bound inside a molecular nanotube. This result is fascinating because the high binding energy of the C60 for the tubular host might have been expected to lead to a high barrier to rotation. The findings from this study will attract the interest of scientists from a wide range of disciplines and the work is significant enough to justify publication in Nature Communications.

The concept behind this study is similar to that of reference 16 (a PNAS paper from the same team), but this new study goes far beyond the previous one. The measurement of rotational dynamics over a wide range of temperatures is a significant advance.

The manuscript is well written and highly readable. It is suitable for publication with minor revision.

We would like to thank this reviewer for his/her time spared for evaluation and also for his/her favourable evaluation. We are pleased to learn that this reviewer kindly finds a significant advance with the present study.

When the authors introduce the [4]cyclo-2,8-chrysenylene molecular nanotube host on page 3, it would be worth pointing out that this compound is a single enantiomer, otherwise readers may assume that it is racemic, then become confused by the discussion of chirality later in the article.

We thank this reviewer for this suggestion. Accordingly, we now describe the single-handedness of our cylinder with an additional reference for this subject. With this revision, we believe that the readers should fully appreciate the unique stereochemistry of the present system.

When the authors state that the rotation frequency of C60 in the host at 335 K is 213 GHz they should state the rotation frequency of C60 in pure C60 at the same temperature, so as to place this value in context.

We agree and provide reference data of pure C60 from the literature.

One very minor suggestion for rewording on page 4, line 85: Change “minute, small” to “minute” or “small”.

We revised this point.